# Rephrase, Augment, Reason: Visual Grounding of Questions for Vision-Language Models

**Archiki Prasad**    **Elias Stengel-Eskin**    **Mohit Bansal**
Department of Computer Science
University of North Carolina at Chapel Hill
{archiki, esteng, mbansal}@cs.unc.edu

## Abstract

An increasing number of vision-language tasks can be handled with little to no training, i.e., in a zero and few-shot manner, by marrying large language models (LLMs) to vision encoders, resulting in large vision-language models (LVLMs). While this has huge upsides, such as not requiring training data or custom architectures, how an input is presented to an LVLM can have a major impact on zero-shot model performance. In particular, inputs phrased in an *underspecified* way can result in incorrect answers due to factors like missing visual information, complex implicit reasoning, or linguistic ambiguity. Therefore, adding visually-grounded information to the input as a preemptive clarification should improve model performance by reducing underspecification, e.g., by localizing objects and disambiguating references. Similarly, in the VQA setting, changing the way questions are framed can make them easier for models to answer. To this end, we present **Rep**hrase, **A**ugment and **Re**ason (REPARE), a gradient-free framework that extracts salient details about the image using the underlying LVLM as a captioner and reasoner, in order to propose modifications to the original question. We then use the LVLM's confidence over a generated answer as an unsupervised scoring function to select the rephrased question most likely to improve zero-shot performance. Focusing on three visual question answering tasks, we show that REPARE can result in a $3.85\%$ (absolute) increase in zero-shot accuracy on VQAv2, $6.41\%$, and $7.94\%$ points increase on A-OKVQA, and VizWiz respectively. Additionally, we find that using gold answers for oracle question candidate selection achieves a substantial gain in VQA accuracy by up to $14.41\%$. Through extensive analysis, we demonstrate that outputs from REPARE increase syntactic complexity, and effectively utilize vision-language interaction and the frozen LLM. [1]

## 1 Introduction and Motivation

Recent advancements in foundational vision-language (VL) models such as GPT-4 (OpenAI, 2023), BLIP-2 (Li et al., 2023), and Flamingo (Alayrac et al., 2022) have enabled tremendous strides in visual understanding tasks (Gan et al., 2022; Zhang et al., 2023a; Yin et al., 2023). Similar to large language models (LLMs) in the text domain (Ouyang et al., 2022; Chowdhery et al., 2022; Touvron et al., 2023, *inter alia*), these large vision-language models (LVLMs) can be guided through well-designed input prompts to perform tasks without fine-tuning, i.e., in a zero- and few-shot fashion. This is a powerful capability, allowing models to be applied to vision-language tasks without access to large annotated training datasets. In this setting, the prompt's phrasing becomes crucial to model performance (Webson & Pavlick, 2021; Mishra et al., 2022; Prasad et al., 2023). Further contributing to the challenge of zero-shot tasks is *underspecification*, a common phenomenon in various VL tasks. In this work, we use visual question answering (VQA) as a representative VL task and seek to improve zero-shot model performance by addressing underspecification. In VQA, underspecified questions might provide inadequate information for an interlocutor to understand their intended meanings and answer them correctly (Pezzelle, 2023; Zhu et al., 2023a; Hu et al., 2022).

---

[1]Our code is puplicly available: https://github.com/archiki/RepARe

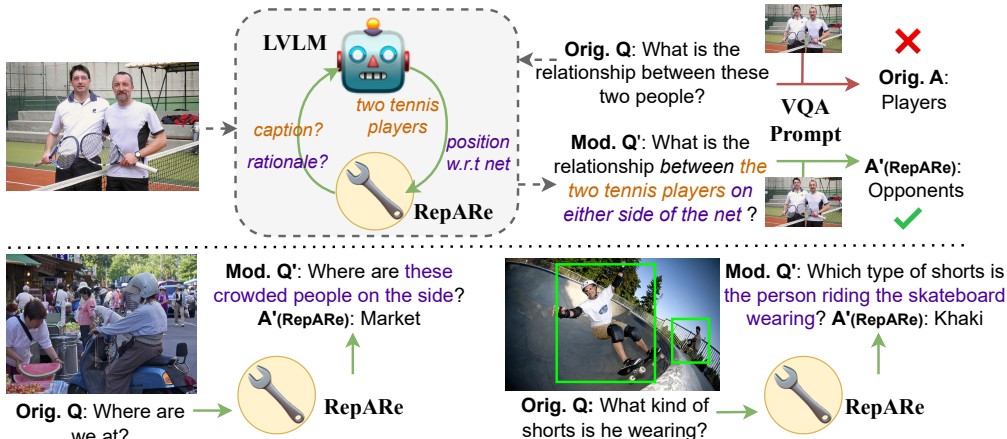

Figure 1: **Top**: The original question (in A-OKVQA) lacks information about implicit reasoning, leading to an incorrect answer. REPARE interacts with the LVLM to extract attributes like "tennis players" and "position w.r.t net" that are key to answering the question correctly. Adding these modifiers to the question elicits the correct response from LVLM. **Bottom**: Underspecified questions from A-OKVQA (left) and VQAv2 (right) datasets along with REPARE outputs.

Underspecification in VL tasks like VQA can manifest in several ways, leading to incorrect model predictions. Firstly, language lacking in visual details (i.e., questions *underspecified w.r.t. image*) can make it harder for models to align text and visual features (Pezzelle, 2023). For example, in Fig. 1 (bottom-left), *"we"* is not grounded to the image. Furthermore, abstract questions often require complex reasoning or external world knowledge that may not be present in the model or at least may be hard to access; in other words, the question is *underspecified w.r.t the world* (Marino et al., 2019; Schwenk et al., 2022). For instance, in Fig. 1 (top), the spatial arrangement of players on opposite sides suggests they are opponents. Explicitly referencing *"the net"* could help the model access this commonsense knowledge. While LVLMs might still rank *"opponent"* highly in their predictions for the original question, the rephrased question more clearly specifies the intent of the inquiry, leading to *"opponent"* as the generated response. Finally, some questions are inherently ambiguous, with multiple valid answers. Even if the model is capable of generating all possible responses, it is not clear which one is intended (Bhattacharya et al., 2019; Stengel-Eskin et al., 2023), i.e., the question is *underspecified w.r.t. intended meaning*. For example in Fig. 1 (bottom-right) there are two men, which results in multiple possible referents for *"he"*. Building upon prior research in textual question reframing (Dong et al., 2017; Majumder et al., 2021; Pyatkin et al., 2023), we hypothesize that making some of the details needed to answer the question more explicit could improve model performance.

There are several paths to addressing the challenges posed by underspecification. One approach involves additional VL pretraining to better align underspecified text to images as well as to enhance the LVLM's internal world model, enabling it to decifer underspecified questions in human-like ways. However, scaling up VL pretraining can be prohibitively expensive (Alayrac et al., 2022; Driess et al., 2023). Note that, in addition to the expense of finetuning, it could be that underspecification in the *training* data leads to continued subpar performance on underspecified questions. Another option is acquiring additional data or information from the user, such as clarifications. This strategy is infeasible for most standard VL benchmarks as they are static datasets (Kiela et al., 2021; Sheng et al., 2021). Furthermore, clarification interactions with users are time-consuming and costly. Thus, our method preemptively incorporates clarifications to reduce ambiguity, emphasize relevant visual details, and suggest reasoning steps, thereby, automatically improving the LVLM's VQA performance without the need for human intervention. Moreover, using preemptive clarifications could also hold value in VL dialogue systems, where users prefer concise interactions but often pose vague questions. This approach has several advantages: (i) it allows for a flexible, gradient-free framework to improve the performance of existing LVLMs without the need for additional pretraining or manual annotations; (ii) our text-based edits are human-readable i.e., we can verify that added details are relevant and consistent with the question's intent; and (iii) crucially, our method harnesses the *asymmetric strength* of most existing LVLMs, whose LLM components typically have far more ca-

pacity and pre-training data than the vision component,[2] and which often have strong reasoning and planning abilities on multimodal data (Wei et al., 2022b; Brohan et al., 2023; Guo et al., 2023). In other words, by preemptively rephrasing questions, we can align more closely with the strengths of existing LVLMs, making rich visual information from the image easier to access.

In Fig. 1 (top), we illustrate at a high level how rephrasing and modifying questions based on the image improves model predictions. Note that our method *does not* have any access to the gold answer, using model confidence to select a question. While the original question elicits a generic response, pinpointing the "*tennis players*" and emphasizing their positions "*relative to the net*" helps the model answer correctly. These modifications are obtained via self-interaction with the LVLM to get more information about the entities in the question as well as other salient objects from model-generated rationales and captions. To this end, we introduce **Rep**hrase, **A**ugment and **Re**ason (REPARE), a gradient-free, instance-level language adaptation framework to address underspecification. Broadly, REPARE consists of two stages: question rephrasing and augmentation, followed by question selection. First, we identify salient entities from the question and generate rationales as well as captions. These features help incorporate visually grounded information into the question. Conditioned on this information, we sample $n$ modified question candidates including the original question. In the next stage, we utilize a confidence-based selection function to choose the most promising candidate, assuming that questions leading to higher-confidence answers are easier for the model to answer, and thus more likely to be correct. The overall pipeline is illustrated in Fig. 2.

Empirically, we show that REPARE improves zero-shot VQA performance by up to 3.85%, 6.41%, and 7.94% on the VQAv2 (Goyal et al., 2017), A-OKVQA (Schwenk et al., 2022), and VizWiz (Gurari et al., 2018) datasets, respectively using LVLMs including BLIP-2 (Li et al., 2023), MiniGPT-4 (Zhu et al., 2023b), and LLaVA-1.5 (Liu et al., 2023a) models in Sec. 4. Note that all percentages we report in this paper are *absolute* improvements. We further demonstrate the capabilities of REPARE in an oracle setting, establishing an upper-bound performance increase of up to 9.84%, 14.41%, and 20.09% on VQAv2, A-OKVQA, and VizWiz tasks, respectively. We extensively evaluate our design choices in Sec. 4.1 and quantitatively show the importance of incorporating visual information to address underspecification, as done in REPARE, compared to paraphrasing in Sec. 4.2. We analyze REPARE's outputs using linguistically-informed metrics like average dependency distance (Gibson et al., 2000) and idea density (Boschi et al., 2017). This reveals that the resulting questions are indeed less underspecified, i.e., more complex (see Sec. 4.3). Finally, in Sec. 4.4, we verify that questions from REPARE make better use of existing LVLMs by leveraging the strength of the LLM while still benefitting from the image. In summary, our contributions include:

- We propose REPARE, a novel zero-shot pipeline that interacts with LVLMs to modify underspecified questions by extracting and fusing information from keywords, rationales, and captions. This grounds questions in the image and commonsense knowledge while also making them less ambiguous, preemptively clarifying them to address underspecification without any human feedback.
- We empirically demonstrate that REPARE boosts zero-shot performance on three standard VQA benchmarks for a collection of LVLMs varying in model architecture, size and VL pretraining by up to 7.94%. Our oracle results suggest that we can obtain as high as 20.09% increase in zero-shot VQA accuracy *solely* by modifying the question.
- Extensive analysis shows that REPARE enhances question complexity via semantic modifications, outperforms paraphrasing, and harnesses LVLM's strengths with simple yet effective modules.

## 2 RELATED WORK

**Large Vision-Language Models.** Significant strides have been made in jointly processing language and images, especially in visual question answering. VQA (Antol et al., 2015; Goyal et al., 2017; Hudson & Manning, 2019; Johnson et al., 2017) has become a benchmark task for VL models. Recent methods address VQA as a zero- and few-shot learning task. These approaches can be categorized into two groups: (i) those relying on continuous image representations (Alayrac et al., 2022; Tsimpoukelli et al., 2021; Zhu et al., 2023b; Liu et al., 2023b; Li et al., 2023, *inter alia*); and (ii) those extracting linguistic information such as captions from images (e.g., Yang et al., 2022;

---

[2]E.g., BLIP-2$_{\text{Flan T5 xl}}$ (Li et al., 2023) consists of a ViT vision encoder, a Q-former fusion module, and a Flan-T5 LLM with 1B, 0.11B, 3B and parameters respectively, i.e., the LLM is ∼3 times more powerful. Note that this is not a permanent feature of such models, and future models could have equally-sized components.

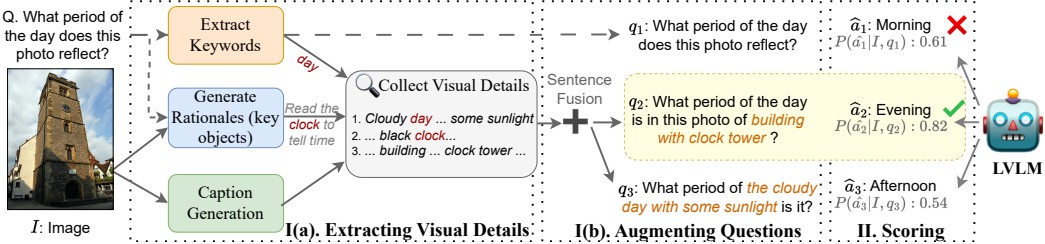

Figure 2: Schematic of REPARE for an image requiring implicit reasoning from A-OKVQA. We first extract keywords, captions, and rationales from the image conditioned on the question, which are used to identify important objects (e.g., day and clock). We query an LVLM about these objects to collect visual details in I(a), that are fused into the original question to produce, in this case, $n = 3$ candidates (I(b)). Lastly, we score and select from candidates using LVLM's answer confidence (II).

Changpinyo et al., 2022; Guo et al., 2023; Berrios et al., 2023). Given the higher performance of projection-based models on VQA, we focus our efforts on the former class.

LLMs can be used for multimodal chain-of-thought (CoT) reasoning (Zhang et al., 2023c); while we use forms of CoT in REPARE, the overall framework differs from CoT. Firstly, CoT is typically open-ended, whereas we follow a principled set of modules, which we validate individually in Sec. 4.1. Secondly, while CoT is generally useful on large models over 100 billion parameters (Magister et al., 2023; Wei et al., 2022a), REPARE can also be applied to LLMs like Flan-T5 (which do not generally benefit from CoT) without any modifications to the model (Wei et al., 2022a).

**Underspecification and Ambiguity.** Underspecification and ambiguity are well-studied within both NLP and linguistics (Schutze, 1995; Futeral et al., 2022; Berzak et al., 2015; Min et al., 2020; Rasmussen & Schuler, 2020). In the multimodal context, Pezzelle (2023) emphasizes underspecification as a significant source of errors in VL tasks – we develop REPARE as a concrete solution to address underspecification by adding visual information. Similarly, Bhattacharya et al. (2019) find underspecification to be a factor contributing to annotator disagreement in VQA, while Stengel-Eskin et al. (2023) focus on ambiguity in VQA and propose a rephrasing method for disambiguation. Unlike REPARE, their method relies on access to gold answers and involves further model training.

**Prompt Editing.** Both LLMs and LVLMs suffer from inherent randomness and sensitivity to choice of training examples, instructions, and prompt template (Zhao et al., 2021; Min et al., 2022; Lu et al., 2022; Awal et al., 2023) in zero-shot and few-shot settings. As a result, several works aim to search for better prompts via gradient-based (Shin et al., 2020; Gao et al., 2021; Jia et al., 2022; Khattak et al., 2023) or gradient-free methods (Sun et al., 2022; Deng et al., 2022; Prasad et al., 2023; Zhang et al., 2023b). However, existing gradient-based methods can be computationally expensive (Sung et al., 2022a), are infeasible for gated models accessible only via APIs, and are often uninterpretable (Khashabi et al., 2022). On the other hand, existing gradient-free methods are primarily designed for language-only models and select the best prompt based on scores (Liu et al., 2022; Prasad et al., 2023) or a learned policy (Deng et al., 2022; Zhang et al., 2023b) using a labeled training set. In contrast, REPARE directly addresses underspecification in VL tasks by making targeted edits to the question using gradient-free, instance-level edits without any train set.

## 3 METHODOLOGY

In this section, we describe the overall pipeline of our method: **Rep**hrase, **A**ugment and **Re**ason (REPARE). Broadly, REPARE consists of two stages: (I) *generating rephrased and augmented question candidates* and (II) *candidate selection*. The first stage yields $n$ modified question candidates, incorporating visual information, and information from rationales using the underlying LVLM. We then use a selection module to identify the best candidate. Note that in all cases, selected questions should preserve the *intent* of the original question while making it easier for the model to answer. Fig. 2 provides a detailed illustration of our REPARE pipeline in action.

### 3.1 GENERATING REPHRASED AND AUGMENTED QUESTION CANDIDATES

**Stage I(a): Extracting Visual Details from Captions and Rationales.** To augment the question with pertinent visually-grounded details, we focus on extracting all relevant information from the image, conditioned on the question.

(i) *Salient Question Entities*: Intuitively, entities mentioned in the question provide vital information about the expected answer. To implement extract key entities from the question, we use an off-the-shelf keyword extraction system (Rose et al., 2010). For instance, it extracts *"day"* from the question in Fig. 2.

(ii) *Information from Rationales*: Answering complex questions can often require world knowledge and implicit reasoning skills (Schwenk et al., 2022). To incorporate this, we sample rationales from the LVLM, which we use to identify relevant objects and features in the image (Chowdhery et al., 2022; Zhang et al., 2023c). This allows REPARE to identify what features might be worth focusing on.[3] For instance, in Fig. 2, the model might extract the clock on the top of the building as an important feature in determining the time of day, based on a rationale like *"Clocks can tell time, so read the clock to determine the time of day."*

(iii) *General Information from Image Captions*: Questions may be underspecified to the extent that they do not contain any salient entities (e.g., *"where are we at"* in Fig. 1). Thus, we also prompt the LVLM to generate a detailed caption for the image. This allows us to capitalize on LVLMs' asymmetric abilities: they excel at image captioning (Alayrac et al., 2022; Tsimpoukelli et al., 2021; Zhu et al., 2023b), and can generate detailed captions (Zhu et al., 2023b; Xie et al., 2022). For example, in Fig. 2, the captioning model might generate a caption like *"A tall, stone building with a clock tower on top on a cloudy day"*.

After identifying salient objects and entities from (i) and (ii), we prompt the LVLM to obtain pertinent details about them based on the image. We add this list to the image captions from (iii) to get the input for for the next stage. We describe the implementation of this stage in detail in Appendix A.3.

**Stage I(b): Rephrasing and Augmenting the Question.** Drawing on work in sentence fusion (Geva et al., 2019; Lebanoff et al., 2020), we leverage the frozen LLM component of the LVLM to incorporate fine-grained details into the question. We combine all the extracted details into a single prompt, and generate $n-1$ modified question candidates, yielding a total of $n$ candidates including the original question (see stage I(b) in Fig. 2). To prevent significant alteration of the question's meaning (especially for yes/no questions), we use an off-the-shelf natural language inference model (Laurer et al., 2022) to discard any candidates that contradict the original question. After generating $n$ question candidates, we prompt the LVLM to answer each question, leading to $n$ question-answer (QA) pairs. We use $n=5$ as default in all our experiments and discuss the impact of increasing $n$ as well as using the full LVLM for sentence fusion in Appendix A.5. Note that all our prompts used within REPARE or for VQA *do not* contain any annotated examples from any VQA dataset (zero-shot setting). Further details and all prompts can be found in Appendix A.2.

## 3.2 QUESTION SELECTION

To select the final QA pair from I(b), REPARE requires a way of scoring the $n$ QA candidates generated using the modules above, in order to choose the QA pair most likely to improve accuracy.

**Stage II: Confidence-based Selection.** As discussed in Sec. 2, most prompt search methods require a labeled dataset to learn a scoring model or a selection policy. In our setting, we perform instance-level edits, meaning that such a supervised scoring scheme would require access to additional annotated data. Therefore, consistent with Liu et al. (2021), at inference time we compute an unsupervised score by utilizing the LLM's ability to self-assess the quality of its generations (Rae et al., 2021; Srivastava et al., 2023; Kadavath et al., 2022).[4] Following Kadavath et al. (2022), we use the LVLM's confidence in generating a proposed answer $\hat{a}_i$ conditioned on the image $I$ and question candidate $q_i$ to select candidate $q'$ (and its corresponding answer $\hat{a}'$) for subsequent evaluation:

$$\text{score}(q_i, \hat{a}_i) = P_{\text{LVLM}}(\hat{a}_i | I, q_i); \quad q', \hat{a}' = \underset{i \in [1,n]}{\operatorname{argmax}}(\text{score}(q_i, \hat{a}_i))$$

**Oracle Setting.** As an upper-bound, we also explore an 'oracle' setting in which we have access to the (gold) annotated answer from the dataset. In this setting, we select the candidate that yields the

---

[3]Note that in the scope of this work, we do not address the veracity and utility of generated rationales which is relatively harder to judge using the same underlying model (Pruthi et al., 2022; Saha et al., 2023). Moreover, for one of the LVLMs (MiniGPT-4) using larger and more powerful LLMs, we prompt the model to generate an explanation in the *common* zero-shot VQA prompt for generating answers. Refer to Appendix A.2 for details.

[4]While past work has found LLMs to be overconfident on a variety of tasks (Mielke et al., 2022; Lin et al., 2022; Zhou et al., 2023; Stengel-Eskin & Van Durme, 2023), this does not impact our results, as we choose $q'$ based on the LLM's *relative* confidence. For more details, we refer readers to Appendix A.5.

correct answer (in case of ties, we perform random selection). This gives us the maximum possible performance of REPARE for a fixed number of candidate questions $n$, discussed further in Sec. 4.

## 3.3 EXPERIMENTAL SETUP

**Vision Language Models.**    We use three recent state-of-the-art LVLMs: BLIP-2 (Li et al., 2023), MiniGPT-4 (Zhu et al., 2023b), and LLaVA-1.5 (Liu et al., 2023a). At a high level, the model architecture comprises of an image encoder (Radford et al., 2021; Fang et al., 2023) and an LLM  (Chung et al., 2022; Chiang et al., 2023) (both frozen) connected by a relatively small trained transformer model called the Q-former (Li et al., 2023). The Q-former acts as a bridge, facilitating information flow between the image encoder and the LLM, resembling an adapter (Houlsby et al., 2019; Sung et al., 2022b).  Beginning with image-to-text pre-training, the Q-former extracts key visual details and then connects to the LLM using a fully-connected layer to project query embeddings into the embedding space of the LLM. Note that while BLIP-2 uses an encoder-decoder-based LLM (Flan-T5), MiniGPT-4 and LLaVA-1.5 use Vicuna with a decoder-only architecture (details in Appendix A.3).

**VQA Datasets and Metrics.**    We use the VQAv2 dataset (Goyal et al., 2017) for general visual understanding. To specifically capture underspecification due to lack of reasoning or world-knowledge, we use the A-OKVQA dataset (Schwenk et al., 2022) containing image-question pairs that require broader commonsense and world knowledge to answer.  A-OKVQA has two settings: (i) directly generating the answer (direct), and (ii) 4-way multiple choice (MC).  Since the test sets of these benchmarks are not publicly available, we report performance on the validation sets (unless mentioned otherwise).  Lastly, we also evaluate on the challenging VizWiz benchmark (Gurari et al., 2018) consisting of real-life information-seeking questions about (often low-quality) images sourced from visually-impaired people.  While developing REPARE, we sample a small set of data points from the train set of the datasets to form our dev set.  In the "direct answer" setting, we use the standard soft VQA evaluation metric for VQAv2, VizWiz, and A-OKVQA (Antol et al., 2015).  In A-OKVQA's MC setting, we use accuracy. See Appendix A.1 for further dataset details.

## 4    RESULTS AND ANALYSIS

In this section, we present the results of our experiments.  First, we establish the effectiveness of the REPARE framework in Sec. 4.1. Then, in Sec. 4.2, we quantitatively distinguish REPARE from simply paraphrasing the question. Furthermore, we provide quantitative analysis of outputs from REPARE, addressing semantic complexity (in Sec. 4.3). Lastly, in Sec. 4.4, we show that REPARE leverages the asymmetric strength of the LLM in an LVLM, allowing the LLM to perform more of the task without eliminating the need for the image.[5]  Note that all improvements in this paper are reported as *absolute* percentage increase.

### 4.1    OVERALL EFFECTIVENESS OF REPARE

**Main Results.**    Our main results are presented in Table 1. When compared to the original questions, using questions after applying REPARE increases the overall zero-shot accuracy of BLIP-2 by $3.85\%$, of MiniGPT-4 by up to $3.02\%$, and that of LLaVA-1.5 by $1.14\%$ on VQAv2. On the A-OKVQA dataset, where answering the question may require a combination of world knowledge and reasoning skills, we show that REPARE improves the zero-shot performance of BLIP-2, MiniGPT-4, and LLaVA-1.5 models by up to $5.47\%$, $6.41\%$, and $3.63\%$ respectively, when directly generating the answer. In the multiple-choice setting with the MiniGPT-4$_{\text{Vicuna 7B}}$ model, this improvement can be as high as $21.54\%$. Moreover, on the challenging VizWiz dataset, REPARE improves performance by $7.94\%$, $3.46\%$, and $2.39\%$ points with MiniGPT-4, BLIP-2, and LLaVA-1.5 models. Furthermore, using gold answers in the oracle setting, we establish empirical upper bounds for REPARE in Table 1. On BLIP-2, REPARE can yield up to $9.84\%$ across both datasets, while using MiniGPT-4, we can obtain a maximum oracle improvement of $14.41\%$ and $33.94\%$ on the A-OKVQA dataset in the direct and multiple-choice settings, respectively. Lastly, REPARE in oracle setting yields up to $7.61\%$ accuracy improvements on the VizWiz dataset. This demonstrates REPARE's efficacy on VQA datasets with different LVLM architectures varying in size and underlying LLM.

**Design Ablations.**    In Sec. 3, we described various design choices made to develop REPARE. In Table 2, we evaluate the effectiveness of different components within REPARE on our dev splits.

---

[5]We refer readers to Sec. 5 for a broader discussion of asymmetric strength and ability.

| Method | VQAv2 | | | | A-OKVQA | | VizWiz |
|---|---|---|---|---|---|---|---|
| | **Overall** | **Y/N** | **Num.** | **Other** | **Direct** | **MC** | **Overall** |
| MiniGPT-4$_{\text{Vicuna 7B}}$ | 51.47 | 71.03 | 27.13 | 42.75 | 27.51 | 41.66 | 29.87 |
| + REPARE | $\underline{54.49}_{\pm 1.44}$ | $\underline{75.77}_{\pm 2.01}$ | $\underline{32.58}_{\pm 2.29}$ | $\underline{43.79}_{\pm 0.72}$ | $\underline{33.23}_{\pm 2.31}$ | $\underline{63.20}_{\pm 5.64}$ | $\underline{37.81}_{\pm 4.26}$ |
| + REPARE (Oracle) | $59.66_{\pm 2.93}$ | $86.56_{\pm 6.12}$ | $38.86_{\pm 4.57}$ | $44.32_{\pm 1.05}$ | $41.92_{\pm 4.86}$ | $75.60_{\pm 7.36}$ | $50.47_{\pm 5.32}$ |
| MiniGPT-4$_{\text{Vicuna 13B}}$ | 61.98 | 82.76 | 39.83 | 51.71 | 41.53 | 56.41 | 44.18 |
| + REPARE | $\underline{64.03}_{\pm 1.26}$ | $\underline{83.54}_{\pm 1.12}$ | $\underline{47.50}_{\pm 3.17}$ | $\underline{53.34}_{\pm 0.98}$ | $\underline{47.94}_{\pm 2.25}$ | $\underline{62.18}_{\pm 4.48}$ | $\underline{51.33}_{\pm 3.79}$ |
| + REPARE (Oracle) | $68.35_{\pm 3.62}$ | $89.33_{\pm 4.28}$ | $51.41_{\pm 5.67}$ | $56.30_{\pm 2.27}$ | $54.20_{\pm 5.18}$ | $80.67_{\pm 8.26}$ | $64.27_{\pm 4.68}$ |
| BLIP-2$_{\text{Flan T5-xl}}$ | 62.58 | 83.99 | 38.63 | 52.91 | 41.86 | 73.89 | 59.63 |
| + REPARE | $\underline{66.43}_{\pm 1.21}$ | $\underline{89.94}_{\pm 3.27}$ | $\underline{48.56}_{\pm 2.85}$ | $\underline{52.94}_{\pm 0.61}$ | $\underline{44.87}_{\pm 1.34}$ | $\underline{77.20}_{\pm 1.45}$ | $\underline{62.38}_{\pm 1.14}$ |
| + REPARE (Oracle) | $72.42_{\pm 3.49}$ | $94.26_{\pm 4.73}$ | $50.67_{\pm 5.06}$ | $61.25_{\pm 2.94}$ | $49.20_{\pm 2.38}$ | $81.14_{\pm 3.61}$ | $65.20_{\pm 2.65}$ |
| BLIP-2$_{\text{Flan T5-xxl}}$ | 65.08 | 85.15 | 39.91 | 56.19 | 41.86 | 76.59 | 62.81 |
| + REPARE | $\underline{68.92}_{\pm 1.36}$ | $\underline{90.54}_{\pm 3.18}$ | $\underline{43.30}_{\pm 2.41}$ | $\underline{58.96}_{\pm 0.87}$ | $\underline{47.33}_{\pm 1.56}$ | $\mathbf{79.23}_{\pm 1.29}$ | $\mathbf{66.27}_{\pm 1.87}$ |
| + REPARE (Oracle) | $74.05_{\pm 3.26}$ | $94.56_{\pm 4.37}$ | $54.40_{\pm 4.72}$ | $63.36_{\pm 2.84}$ | $55.67_{\pm 2.19}$ | $82.80_{\pm 2.36}$ | $70.13_{\pm 2.49}$ |
| LLaVA-1.5$_{\text{Vicuna-7B}}$ | 76.21 | 91.83 | 58.27 | 68.85 | 62.56 | 77.38 | 57.07 |
| + REPARE | $\underline{77.35}_{\pm 0.42}$ | $\mathbf{92.64}_{\pm 0.34}$ | $\mathbf{59.92}_{\pm 0.51}$ | $\mathbf{70.12}_{\pm 0.76}$ | $\mathbf{66.19}_{\pm 1.53}$ | $\underline{78.21}_{\pm 0.37}$ | $\underline{59.46}_{\pm 0.92}$ |
| + REPARE (Oracle) | $79.84_{\pm 1.03}$ | $94.39_{\pm 0.94}$ | $62.07_{\pm 1.25}$ | $73.28_{\pm 1.31}$ | $70.17_{\pm 2.49}$ | $80.75_{\pm 1.27}$ | $62.48_{\pm 1.61}$ |

Table 1: Comparison of baseline zero-shot accuracy (%) and REPARE on VQAv2, A-OKVQA and VizWiz. We run REPARE for $n = 5$ and average performance across 3 random seeds to account for randomness in generating question candidates in Sec. 3.1. We highlight the oracle performance with REPARE using gold answers. The overall best numbers for each dataset are in bold, and the highest numbers for each model are underlined.

- *Importance of Rationales, Captions, and Question Entities*: We measure the utility of details about objects mentioned in the: (i) original question, (ii) image caption, and (iii) rationales, by re-running REPARE with BLIP-2 using *all but one* type of object descriptions. From Table 2, we observe that excluding rationales, captions, or question entities adversely impacts zero-shot performance, with the largest drop in accuracy occurring when rationales are not utilized.
- *Impact of Removing Visual Tokens during Fusion*: Next, we explore the impact of including visual tokens, projected onto the LM, in augmenting the question with visual details in Stage I(b). This involves performing the same sentence fusion task using the entire LVLM, while retaining the image embedding in the input to the frozen LM (refer to Table 2). Our findings reveal that the image embedding can serve as a distraction to the language model when rephrasing the question, resulting in up to a 3.1 point drop in overall accuracy (see qualitative examples in Appendix A.5).
- *Design of Scoring Function*: Lastly, we examine our scoring function described in Sec. 3.2. To ablate the scoring method, we run REPARE but with candidates based on the likelihood of the question *alone*, i.e. $\text{score}(q_i) = P_{\text{LVLM}}(q_i|I)$ instead of $\text{score}(q_i, \hat{a}_i) = P_{\text{LVLM}}(\hat{a}_i|I, q_i)$. Table 2 shows that using question likelihood instead of the answer confidence yields a small drop in the downstream performance by at least $1.09\%$ (further ablations in Appendix A.5).

## 4.2 REPARE ADDS SEMANTIC INFORMATION TO ADDRESS UNDERSPECIFICATION

**Comparison with Paraphrasing in Oracle Setting.** Following past work on leveraging paraphrases to improve QA (Dong et al., 2017), we experiment with a paraphrastic baseline, where we simply paraphrase the question using Pegasus, a strong off-the-shelf model (Zhang et al., 2020).

Table 3 shows that paraphrasing the question leads to major improvements over the zero-shot setting under oracle selection (described in Sec. 3.2). For VQAv2, BLIP-2's performance increases from $62.58\%$ to $70.99\%$ and for A-OKVQA it improves from $73.89\%$ to $79.91\%$ in the multiple choice setting. This indicates that BLIP-2 and its underlying LLM, Flan-T5 are brittle to the phrasing of the question, i.e., without altering the information or meaning of the question, a paraphrased question candidate may yield a higher VQA score (Webson & Pavlick, 2021).

**Comparison with Paraphrasing during Inference.** If the modifications from REPARE were purely cosmetic rewrites, then REPARE and a paraphrastic baseline should have roughly the same performance during inference. Table 3 demonstrates that selecting from paraphrased question candidates without access to gold answers (oracle) presents a challenge. In fact, in 2 out of 3 settings, opting for paraphrased questions results in *lower* performance compared to using the original ques-

| Method | VQAv2 | A-OKVQA |
|---|---|---|
| REPARE | **67.28** | **45.01** |
| w/o Rationales | 64.57 | 43.15 |
| w/o Caption | 66.04 | 44.36 |
| w/o Question Entity | 65.89 | 44.62 |
| w/ $I$ Embeddings in Fusion | 63.18 | 42.38 |
| w/ score $= P_{\text{LVLM}}(q_i|I)$ | 65.47 | 43.92 |

Table 2: Ablation of design choices in REPARE using BLIP-2 on our dev splits (direct answers).

| Method | VQAv2 | A-OKVQA | |
|---|---|---|---|
| | Overall | Direct | MC |
| Baseline (BLIP-2) | 62.58 | 41.86 | 73.89 |
| Paraphrase Oracle | 70.99 | 46.66 | 79.91 |
| REPARE Oracle | **72.42** | **49.24** | **81.14** |
| Paraphrase Selection | 62.91 | 40.23 | 73.57 |
| REPARE Selection | **66.43** | **44.87** | **77.20** |

Table 3: Comparison of REPARE (using BLIP-2) with paraphrasing questions in the oracle setting and unsupervised candidate selection.

tions by up to $1.63\%$. Therefore, although some paraphrased questions may elicit correct answers, choosing them solely based on the model's confidence yields poor results. In contrast, questions generated by REPARE show a distinct pattern: not only do these questions outperform paraphrased questions in the oracle setting, but they are also more easily chosen by the unsupervised scoring function. This indicates that incorporating additional semantic information from both images and rationales in REPARE simultaneously makes questions *easier to answer* as well as *easier to select*.

### 4.3 ANALYSIS OF INCREASED COMPLEXITY IN REPARE'S QUESTIONS

In Sec. 1, we highlight underspecification as a source of errors in VL tasks like VQA. In Table 1, we empirically show that REPARE enhances VQA accuracy across datasets and LVLM architectures. Here, we analyze the questions generated by REPARE and compare them against original questions to confirm that the rephrased questions are in fact more complex, i.e., *less underspecified*. We present quantiative results from two complexity metrics; see Appendix A.4 for qualitative examples.

**Complexity Metrics.** Qualitatively, we find that REPARE questions have increased syntactic and semantic complexity (cf. Table 6). We quantify this with two common complexity metrics: average dependency distance (ADD) and idea density (ID), implemented using BlaBla toolkit (Shivkumar et al., 2020) and Stanza (Qi et al., 2020). *Average Dependency Distance* (ADD) measures the *syntactic complexity* of sentences by calculating the average linear distance between each token and its parent node in a syntactic parse. It is commonly used to measure syntactic complexity (Gibson et al., 2000; Oya, 2011; Liu et al., 2017). ADD ranges on $[0, \inf)$ with a higher score indicating more complexity. *Idea Density* (ID) is the sum of the number of verbs, adjectives, adverbs, prepositions, and conjunctions divided by the total number of words (Boschi et al., 2017). It is commonly used as a measure of *semantic complexity* (Chand et al., 2012; Kemper, 1992). ID ranges between $[0, 1]$ and higher scores indicate more complexity.

| Dataset | Type | ADD | ID |
|---|---|---|---|
| A-OKVQA | Original | 25.40 | 0.282 |
| | REPARE | 32.81 | 0.299 |
| VQAv2 | Original | 17.87 | 0.258 |
| | REPARE | 29.52 | 0.296 |

Table 4: Complexity measures for questions before and after REPARE.

**Quantitative Complexity Analysis.** Our quantitative results can be seen in Table 4, where we compute ADD and ID for a subset of 100 instances from the official validation set. Here, we use BLIP-2 as the backbone for REPARE. Compared to the original questions, both complexity measures are higher for REPARE across models and datasets. This indicates that REPARE adds syntactic complexity and semantic content to the questions; which in turn suggests that the rephrased questions are less underspecified. For example, a REPARE question like *"Why would you use this suitcase packed on both sides?"* from Table 6 has more modifiers than the original, *"Why would you use this bag?"*, leading to a higher ID score. It also has a more complicated syntactic structure, with nested modifiers (*"suitcase packed on both sides"*) leading to a higher ADD.

### 4.4 REPARE LEVERAGES VL INTERACTION TO IMPROVE PERFORMANCE

We further explore the *asymmetric strength* hypothesis (discussed in Sec. 1),[5] which could explain the improvements seen in Table 1. Specifically, we examine how REPARE's addition of visual information to the question allows the LVLM's LLM component to do more of the heavy lifting in the QA task. We test the performance of the original and REPARE questions *without* the image in the input, i.e., to what extent the constituent LLM alone can answer each question cor-

rectly. If REPARE leverages the strength of the LLM well, we should expect the LVLM's LLM-only performance to increase when using REPARE. In Table 5, we evaluate this hypothesis using BLIP-2 as the underlying model. First, we observe that the *image is crucial* to good performance; in all settings, BLIP-2's LLM-only accuracy is quite low. Furthermore, REPARE questions improve in the LLM-only setting, indicating that modified questions take better advantage of the LLM's QA strength (cf. rows 3 and 4). Note the substantial gap of ∼25% between settings with

and without the image for REPARE (cf. rows 2 and 4), which indicates that the rephrased question is *complementary* to the image, i.e., that REPARE does not make questions trivial to answer with just an LLM. Finally, when using just the LLM as the QA model, we find that adding the caption or extracted image details from stage I(a) of REPARE along with the original question improves performance over the original question alone; however, these details do not make up for the lack of the image (c.f. rows 2, 5, and 6 in Table 5). Thus, REPARE improves LVLM performance via both vision-language interaction *and* leveraging the LLM.

| LVLM Setting | VQAv2 | A-OKVQA | |
|---|---|---|---|
| | Overall | Direct | MC |
| [1] Img + Orig. Q | 60.29 | 41.72 | 72.56 |
| [2] Img+ REPARE Q | 67.28 | 45.01 | 78.43 |
| [3] Orig. Q (LLM-only) | 32.84 | 15.93 | 45.20 |
| [4] REPARE Q (LLM-only) | 40.53 | 20.33 | 54.21 |
| [5] Caption + Orig. Q | 52.88 | 27.64 | 65.80 |
| [6] REPARE I(a) + Orig. Q | 54.31 | 30.27 | 66.60 |

Table 5: BLIP-2's LLM-only vs. full model performance on original and rephrased questions.

## 5  DISCUSSION AND CONCLUSION

**Asymmetric Strength and Ability.**   As alluded to in Sec. 1, REPARE is based on two assumptions about existing LVLMs. First, existing LVLMs have larger LLMs than vision components, i.e. they have *asymmetric strength*. Thus, moving some of the burden of the VQA task onto the LLM improves performance, as shown in Sec. 4.4. However, we also assume that the image is still helpful in answering the question – this is also borne out in Sec. 4.4, where visual information still improves the model. This differentiates our work from recent work like Berrios et al. (2023) and Hu et al. (2022) which translate an image into text descriptions and then apply a language-only model for VL tasks, i.e. also make use of asymmetric strength, but not of the image. Our work also holds greater promise for capturing fine-grained visual details, which can be challenging to describe linguistically. Second, REPARE also relies on the asymmetric zero-shot abilities of individual LVLMs. While there is a large gap in QA between zero-shot LVLMs and fine-tuned, task-specific models, LVLMs are competitive at image captioning. We can use this to our advantage, harnessing captions to improve the question. Similarly, while LVLMs may not be able to implicitly reason about the image during QA, their LLMs can extract useful rationales and fuse them into the question.

**Redundancy and Language Bias.**   Qualitatively, much of the information in Table 6 may appear redundant to humans who can perceive the entire image and hone in details already given in the image. It is worth noting that past work has found that humans tend to over-specify when describing visual scenes (Ford & Olson, 1975; Sonnenschein, 1985; Pechmann, 1989; Koolen et al., 2011). In other words, redundancy in descriptions or questions is not uncommon, and may in fact benefit the model. VQA datasets can suffer from language bias, where many questions can be answered correctly without access to the image (Goyal et al., 2017). The analysis in Sec. 4.4 indicates that REPARE questions have stronger LLM-only (i.e., language-only) performance. However, note that the information that gives REPARE questions their higher performance is *extracted from the image* using the same underlying LVLM. Thus, a comparison to a language-only bias here is not entirely accurate, since rephrased questions contain information sourced from the image.

**Limitations.**   One limitation of our method is cost: rather than answering a question directly, we generate several question candidates and then select one. We note, however, that other multi-step approaches, including Chain-of-Thought (Wei et al., 2022b), or exploration-guided reinforcement learning, and search methods also increase the number of tokens and inference steps, and our cost scales linearly in the number of candidates. Note also that while strategies like CoT can help with rationale-style reasoning in particular, they are harder to apply in most existing LVLMs (partly due to the size of LLM components in existing LVLMs, which is typically significantly less than 100B parameters). Addressing underspecification alone is not a cure-all for solving VQA or broader visual understanding tasks. Underlying dataset issues, such as low-quality images and inaccurate human annotations (Bhattacharya et al., 2019), can still prevent models from achieving high accuracy.

ETHICS STATEMENT

Instructions are a useful tool for conveying extrinsic information to LLMs. However, they can also be misused intentionally or unintentionally (Weidinger et al., 2021) in order to alter model outputs to elicit harmful, biased and problematic content. Being based on LLMs, LVLMs are succeptible to similar misuse via targeted instructions or questions. The intended use of REPARE is to obtain modifed questions that work well for LVLMs and help improve model performance for the given instance without significanlty altering the intended meaning; this is orthogonal to whether the original question displays a malicious intent, which is a more general issue that applies to all LLMs/LVLMs. Additionally, in our work we use images and questions from VQA and A-OKVQA; these datasets have been vetted for quality in the past Lin et al. (2014); Goyal et al. (2017); Schwenk et al. (2022) but inappropriate or offensive queries and images could remain since they are quite large. To mitigate the risk of offensive, malicious, or inappropriate questions being generated by REPARE, we manually examined a subsample of 250 generated outputs from REPARE and verified that the generated questions do not display a malicious or offensive intent.

ACKNOWLEDGEMENTS

We thank Jaemin Cho, Peter Hase, Nithin Sivakumaran, David Wan, Jaehong Yoon, and Shoubin Yu for their valuable feedback and inputs for the paper. This work was supported by DARPA ECOLE Program No. HR00112390060, NSF-AI Engage Institute DRL-2112635, DARPA Machine Commonsense (MCS) Grant N66001-19-2-4031, ARO Award W911NF2110220, and ONR Grant N00014-23-1-2356. The views contained in this article are those of the authors and not of the funding agency.

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

# A APPENDIX

## A.1 DATA

We use three datasets, VQAv2 (Goyal et al., 2017), A-OKVQA (Schwenk et al., 2022), and VizWiz (Gurari et al., 2018). VQAv2 is an extension of the original VQA dataset (Antol et al., 2015), which incorporates similar images yielding different answers to the same question. This augmentation doubles the number of image-question pairs, emphasizing the reliance on visual information for accurate answers. While VQA questions are open-ended, the answer vocabulary is relatively limited in size (10M), consisting of mostly one-word responses. In VQAv2, each example is associated with 10 ground-truth answer labels provided by different human annotators. On the other hand, the A-OKVQA dataset is smaller (25K questions in total) but is more challenging. Similar to VQAv2, in the direct answer setting, 10 human annotated 1-2 word answers are provided for each question. The multi-choice setting comes with 4 options along with the index of the correct option. Lastly, the VizWiz dataset contains 32.8K information-seeking questions asked by visually-impaired people based on images clicked on mobile devices. This dataset can be challenging as the images are often blurred, under/over-exposed, or rotated. During the design and analysis of REPARE, we use a separate development set consisting of 5K, 1K, 500 randomly sampled image-question pairs from the train sets of VQAv2, A-OKVQA, and VizWiz respectively. For testing, we use the entire validation set; this corresponds to 214K examples for VQA, 1.1K examples for A-OKVQA, and 4.3K examples for VizWiz. We use the standard VQA metric for open-ended evaluation. According to this metric, a model-generated answer is deemed 100% accurate if at least 3 of the 10 annotators provided that exact answer.

$$\text{Accuracy}_{\text{VQA}} = \min\left(\frac{\text{\# humans that said ans}}{3}, 1\right)$$

The predicted answer is also pre-processed by lowercasing, converting numbers to digits, and removing punctuation/articles. Since LLMs generate free-form text, we constrain the answers using length-penalty of -1 during generation which encourages shorter answers that align better with human annotations.

## A.2 PROMPTS

Table 11 contains an exhaustive list of all prompts used in REPARE for various models. Limited prompt engineering (2-3) trials were done for each prompt on our dev split. For sentence fusion, we use two hypothetical examples (note that these do not include images):

1. **Question**: What is the man wearing?; **Object**: man; **Detail**: he is standing on the sidewalk; **Modified Question**:What is the man who is standing on the sidewalk wearing?
2. **Question**: Are there any flowers?; **Object**: flowers; **Detail**: There is flowers are in a vase. The vase is blue in color and sitting on a table; **Modified Question**: Are there any flowers in the vase on the table?

## A.3 EXPERIMENTAL DETAILS

**Model Checkpoints.** In Sec. 3.3, we use BLIP-2 with ViT-g frozen image encoder (1B parameters), and Flan-T5 XL with 3B model parameters. The pretrained Q-former is an encoder-only transformer model (Vaswani et al., 2017) that shares a similar architecture with BERT (Devlin et al., 2019) comprising of 107M parameters. MiniGPT-4 and LLaVA-1.5 are based on the BLIP-2 architecture with an addition VL pretraining. One key difference is that it uses the Vicuna family of LLMs. We experiment with the two official checkpoints with 7B and 13B model parameters. In Sec. 4.2, we use a popular Pegasus-based paraphrasing model available on HuggingFace (Wolf et al., 2020).[6] We also use an off-the shelf NLI model that achieves competent performance on a suite of NLI benchmarks Laurer et al. (2022).[7] In Sec. 3, we also use the rake_nltk python package.

---

[6]Link to Checkpoint: https://huggingface.co/tuner007/pegasus_paraphrase
[7]Link: https://huggingface.co/MoritzLaurer/DeBERTa-v3-large-mnli-fever-anli-ling-wanli

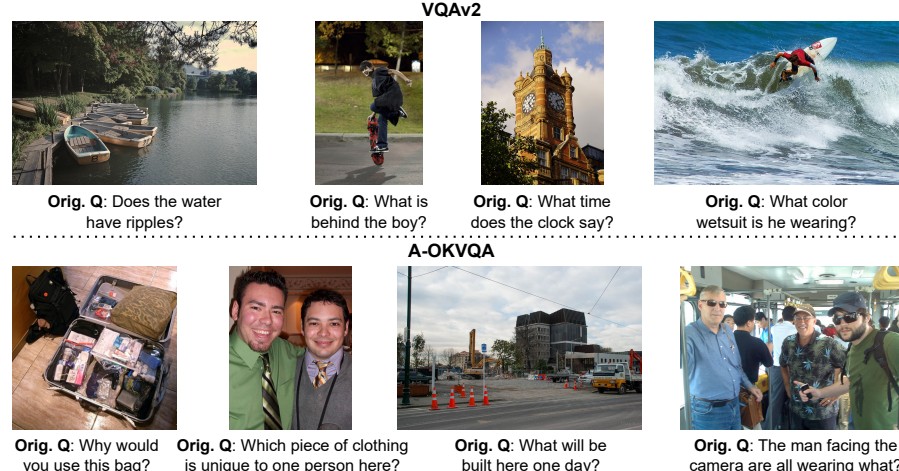

Figure 3: Example images and original questions for Table 6. Some questions (e.g., *"What is behind the boy"* are underspecified, while others refer to small objects in the image (e.g., *"What color wetsuit is he wearing?"*).

**Stage I: Extracting Visual Details and Generating Candidates.** As described in Sec. 3.1, we extract salient visual details from the image using 3 components described below in further detail.

 (i) *Salient Question Entities:* Given only the question from a data instance, we use the keyword extraction tool from `rake_nltk` package to identify salient keywords mentioned in the question. For instance, in the example shown in Fig. 2, "day" is identified as the salient entity.

 (ii) *Information from Rationales:* For this module, we use both the image and the original question for the data point and adopt a two step approach. First, we ask the model to generate an explanation for its answer. Next, based on the explanation and question, we ask the model to identify salient entities mentioned or used in the rationales. Refer to `Rationale (ii)` prompts listed in Table 11.

 (iii) *Information from Captions:* We adopt the straightforward approach of using the `caption` prompts from Table 11 to generate the image caption using the LVLM.

We extract visual information about entities identified in steps (i) and (ii) using the LVLM by querying it using the `extraction of details` prompt listed in Table 11 for each entity separately. This information is concatenated in a prompt in the form of a bulleted list of format `[entity] : [details]`. To this prompt, we add the generic details from the image caption via an additional line: `image: [caption]`. This list of details along with the original question are added to the sentence fusion prompt from Table 11 along with two in-context examples mentioned in Appendix A.2 above which generates the modified question candidates by sampling multiple outputs (same LVLM call).

**Text Generation in REPARE.** To ensure the paraphrasing model generates a valid question ending with '?' we employ constrained decoding by setting a positive constraint on generating the '?' token (Post & Vilar, 2018; Hu et al., 2019). To ensure diverse samples in the sentence fusion stage (determines the diversity of question candidates) we use top-p sampling Holtzman et al. (2019) with $p = 0.95$. To sample rationales, we employ beam search with 5 beams and a temperature of 0.7. After generating question candidates, we filter out sentences that are not valid (do not end with a question mark) or are a verbatim repetition of the original question. Additionally, we also filter out contradictory generations as described in Sec. 3. We sample enough candidates such that we are left with $n$ distinct candidates in the end. If this is not feasible, we repeat the original question in the candidate set to make up for the difference.

**Candidate Selection via REPARE's Score (Stage II).** We employ the VQA prompts mentioned in Table 11 to obtain answers for each question candidate. Typically, the answers (direct answers or option label) correspond to one word or token. In case where we are scoring multiple tokens (as in Sec. 3.2 or Sec. 4.1), we compute the length (number of tokens) normalized log-probabilities that are subsequently exponentiated to obtain probabilites. Note that we are only interested in the relative order, therefore, we can alternatively use log-probabilities to score and select candidates too.

## A.4 Qualitative Examples and Analysis

REPARE questions exhibit an increased degree of specificity, with additional modifiers and fewer ambiguous references, e.g., *"the person riding the wave on the surfboard"* as opposed to *"he"* in the original. Even when questions are unambiguous, REPARE questions include reasoning and location clues. For example, a rephrased question like *"What time is on the clock at the top building?"* indicates which region of the image is important.

| | Original | REPARE |
|---|---|---|
| **VQA** | Does the water have ripples? | Does the water have the small ripples around the boats? |
| | What time does the clock say? | What time is on the clock at the top of building? |
| | What is behind the boy? | What is behind the boy doing a trick on a skateboard? |
| | What color wetsuit is he wearing? | What color is the wetsuit of the person riding the wave on the surfboard? |
| **A-OKVQA** | Why would you use this bag? | Why would you use this suitcase packed on both sides? |
| | Which piece of clothing is unique to one person here? | What piece of clothing is unique to the man on the right? |
| | What will be built here one day? | What will be built at this construction site? |
| | The men facing the camera are all wearing what? | The men facing the camera are all wearing the same pair of sunglasses? |

Table 6: Qualitative examples of original and REPARE generated questions for both datasets with BLIP-2 as the underlying model. For corresponding images, refer to Fig. 3.

Fig. 3 shows the images corresponding to the examples given in Table 6. Each image is paired with its original question as well as the rephrased question from REPARE.

**Answers in Questions.** In some cases, e.g. in the final A-OKVQA question about sunglasses in Table 6, the correct answer (*"sunglasses"*) is added to the question by REPARE. We first note that this is not an unfair advantage, since REPARE operates on the same information as the QA model (image and question), using the same LVLM. Any additional information in a REPARE question is extracted from captions and rationales, which are obtained in a realistic zero-shot test-time setting without any access to the gold answer. Similarly, REPARE's selection module does not use the gold answer in selection. Nevertheless, we report the percentage of times the correct answer is found in the REPARE question and not in the original question. Here, we use the A-OKVQA open-ended (direct) setting and the BLIP-2 model. In a random sample of 100 examples , we find that 7% of rewritten questions from REPARE contain a gold answer. This indicates that part of REPARE's advantage likely comes from an ability to extract the correct answer from the caption and rationale information, incorporate into a question candidate, and then select that candidate.

## A.5 Additional Ablations

**Impact of LVLMs on Generating Question Candidates.** As mentioned in Sec. 3.1, we only used the underlying LLM to fuse or incorporate the extracted visual details into the given question. In Table 2 of Sec. 4.1, we quantitatively show that including the visual tokens, i.e., using the entire LVLM negatively impacts the overall REPARE pipeline and decreases downstream performance. To provide additional insights, Table 7 contains qualitative examples of fusion using only the LLM and the entire LVLM using the BLIP model. We observe that the image embeddings serve as a distraction to the LLM when performing a primarily linguistic task and the resultant question is often ill-formed and heavily dominated by the image caption and/or visual details.

**Alternate ways of computing Answer Confidence.** Kadavath et al. (2022) demonstrate that the self-evaluation ability of model is better in multiple-choice settings than settings in which the LM is required to directly generate the answer. Note that a multiple choice setting is also better specified, since the model is conveyed a set of options to choose from. For instance, if multiple plausible answers exist, only one would be mentioned in the options, indirectly communicating the type of intended response. This is reflected by the contrast in A-OKVQA accuracy (cf. Table 1) in direct and MC settings. In the direct answer setting, we compare computing model's answer confidence in

| | Original | REPARE w/ LLM Fusion | REPARE w/ LVLM Fusion |
|---|---|---|---|
| VQA | Does the water have ripples? | Does the water have the small ripples around the boats? | Yes, there are ripples in the body of water where a group of boats are docked? |
| | What time does the clock say? | What time is on the clock at the top of building? | What time does the clock tower say in the building? |
| | What is behind the boy? | What is behind the boy doing a trick on a skateboard? | A person is doing a trick on a skateboard behind the boy in the middle of a parking lot in the city at night in front of a large building with a skyscraper in the background? |
| | What color wetsuit is he wearing? | What color is the wetsuit of the person riding the wave on the surfboard? | What color wetsuit is he riding in? |
| A-OKVQA | Why would you use this bag? | Why would you use this suitcase packed on both sides? | Why would you use this bag with a lot of items in it? |
| | Which piece of clothing is unique to one person here? | What piece of clothing is unique to the man on the right? | What piece of clothing is unique to one person: a man and a woman posing for a picture? |
| | What will be built here one day? | What will be built at this construction site? | A truck is driving down a street with construction cones and a construction site in the background, what will be built there one day? |
| | The men facing the camera are all wearing what? | The men facing the camera are all wearing the same pair of sunglasses? | The men facing the camera are all wearing the same thing, what is it? |

Table 7: Qualitative comparison of generated question candidates with and without visual tokens in Stage II (sentence fusion) of REPARE. Corresponding images shown in Fig. 3.

two additional ways that Kadavath et al. show to be better calibrated. First, we compute True/False answer confidence by adding the following suffix to the VQA prompt:

```
Proposed Answer: [$\hat{a}_i$].
Is the proposed answer true or false? (A) True, (B) False.
The proposed answer is:
```

Then, we use $P_{\text{LVLM}}(\text{True})$ as a substitute for answer confidence $P_{\text{LVLM}}(\hat{a}_i | I, q_i)$. Additionally, we also employ the strategy of showing the model multiple generated answers to estimate answer confidence. For this we take advantage of $n$ different question candidates that yield different answers $\{\hat{a}_i\}_{i \in [1,n]}$. Hence, we add the following prefix to the VQA prompt:

```
Plausible Answers: [$\{\hat{a}_i\}_{i \in [1,n]}$].
Proposed Answer: $\hat{a}_i$.
Is the proposed answer true or false? (A) True, (B) False.
The proposed answer is:
```

We denote this setting as $P_{\text{LVLM}}(\text{True} | \{\hat{a}_i\}_{i \in [1,n]})$ and substitute this probability instead in the score function. The results are shown in Table 8. We find that all the implementations of LVLM's answer confidence yield comparable performance across datasets (<1 point difference). Since we use the same LVLM to rank various question candidates for a given image, the relative ordering of scores (Sec. 3.2) should not be significantly affected by the model's calibration. Post-hoc calibration methods such as Platt scaling (Platt et al., 1999) or isotonic regression (Zadrozny & Elkan, 2002) preserve relative ordering, so they would have no effect on the selection criterion.

| REPARE score | VQA | A-OKVQA |
|---|---|---|
| $P_{\text{LVLM}}(\hat{a}_i | I, q_i)$ | 67.28 | 45.01 |
| $P_{\text{LVLM}}(\text{True})$ | 68.01 | 44.78 |
| $P_{\text{LVLM}}(\text{True} | \{\hat{a}_i\}_{i \in [1,n]})$ | 67.56 | 44.87 |

Table 8: Comparison of performance of REPARE with BLIP-2 using different score functions for computing answer confidence.

**Imapct of increasing the number of candidates $n$ in REPARE.** In Fig. 4, we explore the impact of increasing the number of candidates in REPARE, i.e., $n$ on its effectiveness at enhancing BLIP-2's VQA (direct) performance. We find that initially increasing from $n = 2$ to $n = 5$ leads to performance gains in both the inference and oracle settings for VQAv2 and A-OKVQA datasets. However, the gains saturate after $n = 10, 15$ for both datasets. In fact, during inference, wherein we select 1 out of $n$ question candidates, we find the VQA accuracy gradually decreases

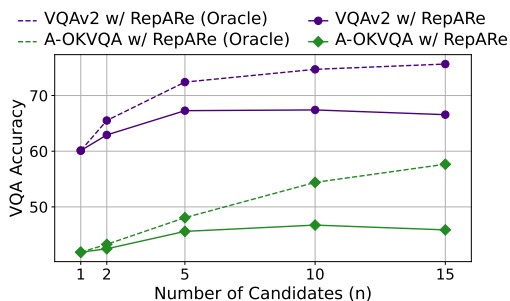

Figure 4: Trends in VQA performance of REPARE for different values of $n$.

at $n = 15$. This is expected, since increasing $n$ allows for diverse candidates; however, selection from a very large pool of candidates (like $n = 15$) is more challenging, and REPARE's selection module is more likely to make a suboptimal choice – hence the growing gap between oracle and REPARE performance.

**Analysis with MiniGPT-4.** In Sec. 4, we use BLIP-2 for conducting our analysis. However, BLIP-2 uses an encoder-decoder LLM while the remaining LVLMs (MiniGPT-4 and LLaVA-1.5) use Vicuna, which is a decoder-only LLM. In this section, we show that the choice of underlying LLM architecture does not impact the relative trends. In Tables 9 and 10, we repeat the analysis in Sec. 4.1 with MiniGPT-4 models corresponding to Tables 2 and 3 respectively. Our ablation study in Table 9 once again highlights the importance of each component of RepARe in improving VQA performance. Similarly, Table 10 reveals that even with MiniGPT-4 as the underlying LVLM, candidates generated by REPARE significantly outperform paraphrased question candidates when selected based on the model's answer confidence.

| Method | VQAv2 | A-OKVQA |
|---|---|---|
| REPARE | **57.74** | **31.23** |
| w/o Rationales | 53.29 | 28.62 |
| w/o Caption | 56.49 | 29.51 |
| w/o Question Entity | 54.91 | 29.28 |
| w/ $I$ Embeddings in Fusion | 54.49 | 28.97 |
| w/ score $= P_{\text{LVLM}}(q_i|I)$ | 56.46 | 29.19 |

Table 9: Ablation of design choices in RE-PARE using MiniGPT-4$_{\text{Vicuna 7B}}$ on our dev splits (direct answers).

| Method | VQAv2 | A-OKVQA | |
|---|---|---|---|
| | Overall | Direct | MC |
| Baseline (MiniGPT-4) | 51.47 | 27.51 | 41.66 |
| Paraphrase Oracle | 57.41 | 39.83 | 73.68 |
| REPARE Oracle | **59.66** | **41.92** | **75.60** |
| Paraphrase Selection | 51.39 | 26.28 | 40.52 |
| REPARE Selection | **54.49** | **33.23** | **63.20** |

Table 10: Comparison of REPARE (MiniGPT-4$_{\text{Vicuna 7B}}$) with paraphrasing questions in the oracle setting and unsupervised candidate selection.

| | Dataset | Setting | Prompt |
|---|---|---|---|
| **BLIP-2** | VQAv2 | VQA Prompt | `Question: [Question] Short Answer:` |
| | A-OKVQA (MC) | | `Question: [Question]`
`Options: A. [Choice 1], B. [Choice 2], C. [Choice 3] , D. [Choice 4]`
`Answer: Option` |
| | | Caption | (Default, empty string) |
| | | Rationale (i) | `[VQA Prompt] Explanation:` |
| | All | Rationale (ii) | `[LVLM Response for Rationale (i)]`
`Question: [Question]`
`Which all entities or objects from this image would I need to observe to answer this question?` |
| | | Extraction of Details | `Question: What can you tell me about [entity] in this image?` |
| **MiniGPT-4** | VQAv2 | VQA Prompt | `### Human:  <ImageHere> </Img>### Human: Based on the image, answer the question below in preferably only 1 word.`
`Question: [Question]` |
| | A-OKVQA | VQA Prompt (i) | `### Human:  <ImageHere> </Img>### Human: Based on the image, answer the question below. Explain your answer.`
`Question: [Question]` |
| | | VQA Prompt (ii) | `[VQA Prompt (i)]### Assistant: [LVLM Response]`
`### Human: Shorten your answer to the question as much as possible, preferrably only 1 word.` |
| | A-OKVQA (MC) | VQA Prompt (i) | `### Human: Based on the image, select the correct answer to the question from the options. You MUST mention option labels, i.e., 'A.', 'B.', 'C.' or 'D.' in your response. Explain your answer.`
`Question: [Question]`
`Options: A. [Choice 1], B. [Choice 2], C. [Choice 3] , D. [Choice 4]` |
| | | VQA Prompt (ii) | `[VQA Prompt (i)]### Assistant: [LVLM Response]`
`### Human: So which option is your final answer: 'A.', 'B.', 'C.' or 'D.'?` |
| | All | Caption | `### Human:  <ImageHere> </Img>### Human: Describe the image in a couple of sentences.` |
| | All | Extraction of Details | `### Human: What can you tell me about [entity] in this image?` |
| **LLM** | All | Rationale (ii) MiniGPT-4 | `### Human: You are given a description of an image, a question and its response below.`
`Image Content: [Caption Response]`
`Question: [Question]`
`Response: [Rationale Response from VQA prompt].`
`List up to 3 objects or from the image were relevant to answering the question? Describe each object ONLY 2-3 words.### Assistant: Enumerated list of top-3 relevant objects used:` |
| | | Sentence Fusion[†] | `You are given a question about an image. Modify the question by adding descreptive phrases to entities based on the provided details. Both original and modified questions MUST have similar meaning and answer. [2 Hypothetical Examples]`
`Question: [Question]`
`Details: [Bulleted list of entities and 1-2 sentences of corresponding details.]`
`Modified Question:` |

Table 11: All the prompts used in REPARE. '(i)' and '(ii)' indicate a sequential conversation. [†]For Vicuna model, we add ### Human, and ### Assistant prefixes.

