# OpenReview forum: "Rephrase, Augment, Reason: Visual Grounding of Questions for Vision-Language Models"
_ICLR.cc/2024/Conference — ICLR 2024 poster_

### Official Review · Reviewer_KySW · 2023-10-31

**Soundness:** 3 good
**Presentation:** 3 good
**Contribution:** 2 fair
**Rating:** 6
**Confidence:** 4

**Summary:**

This paper proposed a pipeline framework for improving the accuracy of LVLMs in VQA tasks.
First, details are identified from the question and the image, including entities (using off-the-shelf tools) and rationales (using the LVLM), which are then used to obtain related details from the image (using the LVLM), and descriptions from the image (using the LVLM).
Second, the details are fused with the question, generating multiple question candidates (using the LVLM), the results of which are filtered to exclude semantically inconsistent ones (using off-the-shelf tools).
Finally, the LVLM answers each candidate question; the final answer is selected using the LVLM with the confidence-based method.

On two benchmark sets, i.e., VQAv2 and A-OKVQA, solid improvements are shown on 3 LVLMs. The with/without kind of ablation is conducted suggesting each element in the pipeline contributes to the improvement. Further analyses are conducted to try to show that the new questions are effective because of less underspecification and ambiguity, i.e., because of the added details that are obvious for humans but not quite so for LVLMs. The asymmetric strength hypothesis seems to suggest that the visual components alone are not quite up to the job of the VQA task.

**Strengths:**

- The motivation is straight-forward, clear, and reasonable.
- The improvements seem solid and the analyses support the improvements that come from the proposed pipeline.
- The code and the data are provided and the authors promised public release. This is very important because the proposed pipeline is complicated and not easily reproducible.

**Weaknesses:**

- The pipeline seems overcomplicated and involves many steps that are indispensable to the overall performance. Unlike plain CoT, which usually conducts inference once, the proposed pipeline conducts inference multiple times using the LVLM and involves off-the-shelf tools twice. The complexity may affect the reproduction of the method and the incurred (computation and time) cost may hinder the adoption of the method in application.
- The main results (Table 1) need more explanation. (1) For example, the standard deviation considering the oracle implementation is high. I did not expect that using the optimal candidate would lead to higher variance. Are there results regarding the choice of the number of the question candidates? (2) I would love to see more QA datasets (from diverse sources) tested on. (3) I wonder if the asymmetric strength hypothesis holds, is it possible that stage 2 and stage 3 can be changed to using the original question and the extracted details in texts without the image? (4) All analyses are based on BLIP-2, an encoder-decoder model. I don't think I find a discussion on the effect of model architecture (encoder-decoder or decoder-only).
- The writing and the organization can be improved. Personally, I would like Section 3 Methodology to be more straightforward. From what I understand, Stage I (ii) adopts different post-processing from (i) and (iii). The paragraph before Section 3.2 states "we prompt the LVLM to answer each question" and the first paragraph in Section 3.2 states "To select which question to answer", which are contradictory. I had to check the appendix and the footnotes multiple times to guess what's going on.

**Questions:**

Please see the numbered points in weaknesses and comments.

---

> ### Author Response · Authors · 2023-11-18
> **Response to Reviewer KySW (Weaknesses and Clarifications)**
>
> We thank you for a detailed review of our paper and helpful suggestions as well as highlighting our “solid” improvements and supporting analysis. We address your comments below.
>
> **On complexity of our pipeline**
> * **Applicability of CoT:** While our system contains multiple components, we demonstrate through our ablations that (as you point out) each component is necessary. We experimented with CoT approaches, finding that LVLMs generally did not benefit. This is in line with prior work highlighting the limited applicability of in-context learning ([Li et al. 2023](https://arxiv.org/pdf/2301.12597.pdf), Section 5 (1st paragraph) and, [Huang et al. 2023](https://arxiv.org/pdf/2302.14045.pdf), Table 13 shows little to no improvement from CoT across multiple tasks). In our setting, in-context learning is a required ability for CoT, since we need to give the model examples of the types of CoTs we would like it to generate (as is common practice, e.g. [Lu et al. 2023](https://arxiv.org/pdf/2209.09513.pdf), Fig. 5). In our case, the addition of multiple images into the context led the model to answer questions about the in-context images, rather than the test image.
> * **Comparison of token usage of RepARe with CoT:** Disregarding poor performance of CoT (as discussed in Sec 2 paragraph 2 on LVLMs of our original submission), we have additionally collected estimates for the number of tokens shown to models with RePARe and with CoT, using a reasonable number of demonstrations (8). Here, RepARe requires between 430-550 LVLM tokens per task instance on an average, while CoT requires ~460 LVLM tokens on average. Given that some of the modules in RepARe can be parallelized, we contend that RepARe is comparable to CoT with in-context learning in terms of efficiency (measured in LVLM tokens) while having the added benefit of strong performance. We note that the computational cost of the off-the-shelf tools we use is negligible compared to the LVLM (which would be shared by CoT approaches). We also add this clarification to Sec 5’s limitations paragraph.
>
> **Addressing writing clarity**
> We have updated Sec 3 to improve clarity, focusing on the areas you have mentioned. First, we have split the stages up differently (see new Fig. 2). For 3.1, we have clarified what the output from the stage is. We have reordered “salient question entities” and “information from rationales” to be grouped together to improve clarity, and have added details in the last paragraph under “Stage I(a)”. We have also added clarification to Stage I(b) and Section 3.2 to indicate that what we are scoring in 3.2 is in fact question-answer pairs, not just questions or answers. Additionally, we provide an exhaustive list of prompts used in this work and describe their usage within each module in detail in updated A.3 for further clarification. We hope that this addresses any lack of clarity in the prior version of the section.

---

> > ### Author Response · Authors · 2023-11-18
> > **Response to Reviewer KySW (Numbered Questions)**
> >
> > 1. We understand the possible confusion here. As originally mentioned in Section 3.2, there is inherent randomness in the process of generating question candidates. Apart from the original question, the other candidates may not overlap at all between any 2 random runs. This is independent of the selection strategy (answer confidence or using gold answers). However, when we use gold answers for selection in the oracle setting, even peculiarly phrased questions that might yield the correct answer but with low LVLM confidence can be selected which increases the standard deviation (when this candidate is absent in subsequent runs). We have clarified this the updated caption for Table 1. On your second point, Fig. 4 in updated Appendix A.5 shows how RepARe’s performance scales with the number of candidates.
> > 2. Following your suggestion, in updated Table 1, we have now added an additional dataset, VizWiz, along with additional models – LLaVA-1.5 and BLIP-2 XXL. In all these settings, we show that RepARe effectively increases VQA accuracy by up to 7.94% points.
> > 3. Following your advice, we have added rows 5, and 6 to updated Table 5, testing this hypothesis.  Row 5 shows that adding the caption improves LLM QA performance instead of just using the question but does not match full LVLM accuracy (with the image). We added the raw textual details extracted from stage 1(a) with the original question to directly generate the answer (without the image) and find that while our extracted details are more beneficial than captions alone, they lag behind LVLM baseline using the image and the original question by nearly 6% and trail the LVLM with image + RepARe question by at least 12%.
> > 4. We include Tables 9 and 10 (corresponding to Tables 2, and 3 in the original submission) with the MiniGPT4 model in the updated Appendix A.5 and find all the trends hold irrespective of underlying encoder-decoder or decoder-only architectures of the LVLM.
> >
> > Please let us know if you have any additional questions or suggestions. Thanks again for your detailed feedback.

---

> > > ### Author Response · Authors · 2023-11-21
> > > **Follow-up on rebuttal**
> > >
> > > Dear reviewer, thanks again for your detailed review.
> > >
> > > Since we are nearing the end of the discussion period tomorrow (*Nov 22, 2023*), we wanted to politely reach out to see if our response has addressed your questions satisfactorily (and we would greatly appreciate it if you could revisit your scores accordingly). We are also happy to answer any other/follow up questions before the deadline. Thanks again!

---

> > > > ### Comment · Reviewer_KySW · 2023-11-22
> > > > **Response to Rebuttal**
> > > >
> > > > I thank the efforts made by the authors for rebuttal. The rebuttal generally solves my concerns. I raise my score to 6.

---

### Official Review · Reviewer_j2wD · 2023-11-01

**Soundness:** 3 good
**Presentation:** 2 fair
**Contribution:** 2 fair
**Rating:** 6
**Confidence:** 3

**Summary:**

The paper proposes an pipeline for using LVLMs to solve VQA that modifies the question using visual information from the image and select the answer with the best confidence. Experiment results on VQAc2 and A-OKVQA show improvement in all question categories. The authors use ablation studies and shown that the visual details and question entity do help improve the performance.

**Strengths:**

1. The performance improvement from the method seem to be solid.
2. The paper is mostly clear with extensive experiments.

**Weaknesses:**

1. Some of the text requires further clarification.
2. The underlying hypothesis should be stated more clearly, which in my understanding is that distribution of more specified questions are more aligned with the training data of LVLMs and the answers with higher confidence are more likely to be correct.

**Questions:**

1. Any explanations on why questions with numeric answers benefit the most from RepARe?
2. What does it mean using `golden` answers for selection? And also `paraphrase oracle`.
3. In 4.4, why is that "In all cases, REPARE increases the gap between LLM-only and BLIP-2"? As it seems that the gap actually decreases. (The numbers should be negative)
4. The baseline of adding captions alone to the question, i.e. <caption><question>, should be compared.

---

> ### Author Response · Authors · 2023-11-18
> **Response to Reviewer j2wD**
>
> Thank you for highlighting our “solid” performance improvements, “extensive experiments” as well your insightful questions and comments. We include our detailed responses below.
>
> **On underlying assumptions.** Thank you for raising this issue. To clarify, our assumptions/hypotheses are regarding the strengths/weaknesses of models, rather than about the alignment to the pre-training data. Our core assumptions are that
> 1. LVLMs generally have a stronger LLM component than visual encoder (asymmetric strength)
> 2. LVLMs are better at tasks like captioning than QA (asymmetric abilities)
> 3. Finally, we assume that we can use answer confidence to rank answers (and thus, implicitly rank the rewritten questions that generated them).
>
> We see in our results that these assumptions are borne out (updated Table 5, where we see a large boost to the LLM’s performance from RepARe and from using generated captions, and Table 2, where we ablate the scoring method). We have updated our introduction (Sec 1, page 3) as well as Sec 5 to more clearly state this. Additionally, for the third assumption we experiment with alternate ways of computing answer confidence and find that it does not significantly impact downstream task performance of RepARe (Table 8 updated).
>
> **Q1: On numeric improvement.** We qualitatively examined the best candidate questions for numeric examples. We found that correctly-answered questions often include additional localization. For example, rather than answering *“how many teddy bears are in the photo?”* the model would answer *“how many teddy bears are on the table?”*. We hypothesize that this kind of additional localization helps the LVLM focus on the correct image regions, simplifying the overall task.
>
> **Q2: Clarification of terms.** We apologize for the confusion. By *gold answers*, we are referring to the ground-truth (gold) answers contained in the datasets. The oracle setting (in the original Sec 3.2, now highlighted) refers to using the ground-truth answers to select the question candidate, i.e. choosing the question that yields the correct answer. However, such information is not available at test time. Similarly, the *paraphrase oracle* (in original Section 4.2, now highlighted) uses the ground-truth answers to select a candidate from a pool of paraphrases of the original question. This setting is designed to set an upper-bound on improvements from rephrasing/paraphrasing by directly using the ground-truth answer.
>
> **Q3 & Q4: Revisions to Sec 4.4.** We understand the source of your confusion in interpreting Table 5. In the original table, we had endeavored to show that RepARe’s improvements are largely, but not exclusively, coming from the LLM. We have since overhauled the table and the bulk of Sec 4.4 to convey this message more clearly, highlighting the following key points in our updated paper:
>
> * RepARe improves LLM-only performance showing that the modified questions leverage the asymmetric QA strength of the LLM.
> * Despite the improvement in LLM-only performance, there remains a substantial gap in performance with and without the image (~25% points). Therefore, the modified question is *complementary* to the image and does not make answering the question trivial without the image.
> * Following your suggestion, we include a baseline in which the caption and the original question (excluding the image) are given to the LLM and find that while this setting outperforms the question-only baselines, it still trails behind the image + original question and image + RepARe settings. This shows that RepARe incorporates visual information that is *complementary to the question and the caption* (added as row 5 of updated Table 5).
>
> We hope that our changes have satisfactorily addressed your concerns. Please feel free to reach out to us with any additional questions and writing suggestions.

---

> > ### Comment · Reviewer_j2wD · 2023-11-21
> > **Response to rebuttal**
> >
> > Thanks for the detailed response and the new results. I will raise the score to 6. However, I still have doubt on the effectiveness of the proposed method compared to captioning. For the caption baseline, I actually mean the results when picture is included, which is picture+caption+original question.

---

> > > ### Author Response · Authors · 2023-11-21
> > > **Response to follow-up from Reviewer j2wD**
> > >
> > > Thank you for your reply. As for the baseline with captions in Table 5, we understand your suggestion. Based on your follow-up clarification, we just ran another baseline with image + caption + original question on VQA and A-OKVQA, with results shown below:
> > > Setting | VQA | AOKVQA (direct) | AOKVQA (MC)
> > >  :---  | :---: | :---: | :---:
> > > Img + Orig. Q | 60.29 | 41.72 | 72.56
> > > Caption + Orig. Q | 52.88 | 27.64 | 65.80
> > > Img + Caption + Orig. Q | 60.63 | 40.28 | 73.71
> > > Img + RepARe Q | 67.28  | 45.01 | 78.43
> > >
> > > We observe that while including image + caption + orig. Q improves performance over caption + orig. Q (comparing rows 2&3), it does not significantly improve over image + Orig. Q, and more importantly (comparing rows 1&3) ***still trails behind the performance gains from RepARe by ~5%*** (row 4).
> > >
> > > We hope that these results address your remaining questions and feel free to discuss/revisit anything else.

---

### Official Review · Reviewer_Jt1P · 2023-11-09

**Soundness:** 3 good
**Presentation:** 3 good
**Contribution:** 3 good
**Rating:** 6
**Confidence:** 4

**Summary:**

This paper introduces "Repare", a gradient-free framework that consists of three phases: Visual Details Extraction, Question Rephrasing and Augmentation, and Question Selection. "Repare" rephrases questions with more precise information, drawing from image captions, question keywords, and rationale details. Following rephrasing, Repare produces multiple question variations, which are then filterd through an unsupervised quality score provided by a Language Model (LM). The paper concludes with an assessment of Repare's performance on VQAv2 and A-OKQA datasets, showing enhancements when compared to the BLIP2 and MiniGPT4 baseline models.

**Strengths:**

- The paper is easy to read and generally well-written
- Interesting idea of improving VL models in VQA tasks by just modifying one modality (e.g text).
- Improvements over baselines (BLIP2, MiniGPT4) looks reasonable.

**Weaknesses:**

- Evaluation suite should be improved. For example including: TextVQA, VizWiz. Additionally, authors should consider evaluating tasks such as HatefulMemes which might be more challenging to the proposed approach. Also, consider recent evaluation tasks such as MME.
- Including recent instruct multimodal models (e.g. LLaVa, Qwen-VL) would be an interesting experiment to see if the gain with Repare is still relevant in these models.
- Minor:
    - >One approach involves additional VL pretraining ...

         In this case, LENS does not involve additional multimodal pretraining.
  - Why not include BLIP2-XXL as one of your main baselines and improve it with Repare?

**Questions:**

Please take a look at the weaknesses.

---

> ### Author Response · Authors · 2023-11-18
> **Response to Reviewer Jt1P**
>
> Thank you for your detailed feedback and helpful comments. We appreciate that you find our idea of focusing on the text modality to improve VQA performance interesting. In addition to the general response, we have addressed your comments below.
>
> **Improving evaluation suite.** Your comments about expanding the evaluation framework by including more datasets and models are well-taken. Following your suggestion, we have evaluated the VizWiz dataset in Table 1 (updated) and also included performance of LLaVA-1.5 on all 3 datasets in Table 1 (updated) using the Vicuna-7B as the underlying LM. Note that much like TextQA, a considerable portion of VizWiz questions also require the model to utilize the text in the images.  Also taking your feedback into account, we have added BLIP2-XXL to our evaluation.
>
> **Takeaways:** Results in Table 1 (updated) indicate that our method RepARe proves to be effective on VizWiz improving accuracy by up to 7.94% (absolute points) for MiniGPT-4 models and yields 7.61% absolute improvement in accuracy on the A-OKVQA dataset with the LLaVA-1.5 model (with additional tuning experiments running). These improvements show that RepAre is robust to poor image quality, as VizWiz images are often blurred, under/over-exposed, or rotated ([Gurari et al. 2018](https://arxiv.org/pdf/1802.08218.pdf)) since they were collected by visually-impaired people on mobile devices. Lastly, we show that RepARe is also effective at improving performance of BLIP-2 XXL in Table 1 by 3.84%, 5.47%, and 3.46% on VQAv2, A-OKVQA (direct), and VizWiz datasets respectively. In addition to Table 1 (updated), we discuss these promising results in the revised Sec 4.1 (highlighted).
>
> **Addressing “minor” comments.** We understand the confusion caused by citing LENS in the introduction, however our objective was to reiterate LENS’s motivation about additional pre-training costs of Q-former-like models despite frozen image encoders and language models. We have now removed this citation from the sentence in question to avoid confusion.
>
> We hope that these additional experiments address your concerns. Please let us know if you have any follow up questions.

---

> > ### Author Response · Authors · 2023-11-21
> > **Follow-up on rebuttal**
> >
> > We would like to follow up to check whether our response addresses your concerns or if you have any further questions. We would really appreciate the opportunity to discuss this further if our response has not already addressed your concerns. We thank you again for your time and detailed comments.

---

### Author Response · Authors · 2023-11-18
**General Response to Reviewers**

We thank all the reviewers for their extensive reviews and helpful feedback. We are grateful to the reviewers for highlighting "solid improvements", "well-written", "interesting idea of improving VL models", and "extensive experiments". Based on your feedback, we have edited the paper with the following key changes highlighted in blue.

* **Improving the evaluation suite:** Based on feedback from Reviewers Jt1P and KySW, we have expanded the evaluation suite in Table 1 to include an *additional and challenging dataset* VizWiz along with *two additional models* BLIP-2 XXL and LLaVA-1.5, showing improvements with RepARe for new datasets and models by up to 7.94% points.

* **Stating Underlying Assumptions:** Following comments from Reviewer j2wD, we have improved Sec 1 by stating the underlying assumptions with supporting results in Sec 4.4 and Appendix A.5.

* **Presentation of our method:** Following suggestions from Reviewer j2wD and KySW, we have added clarifications to Sec 3 explaining our method clearly and modified Fig 2. To enhance reproducibility of our implementation, we include detailed descriptions and prompts used in our pipeline in Appendix A.3.
* **Asymmetric Strength Hypothesis:**  Based on feedback from Reviewers j2wD and KySW, we have rewritten Sec 4.4 to improve clarity and included additional baselines (rows 5 and 6 of Table 5) to support our hypothesis.

* **Analysis and Ablations:** Lastly we provide additional analysis and ablations for RepARe in Appendix A.5. Note that we have moved the discussion of qualitative examples in Sec 4.3 to the appendix A.4, which has changed the table numbers in the revised pdf. ***In our response we use the revised table numbers based on the updated submission.***

---

### Meta-Review · Area_Chair_QiQm · 2023-12-14

**Metareview:**

While RepARE shows promise for improving VQA performance by modifying the text prompt, reviewers raise concerns about its complexity, limitations of the evaluation protocols, and the need for further clarification on its workings.

However, the work achieves solid performance enhancements over baselines like BLIP2 and MiniGPT4 on various metrics. Modifying the question to be more aligned with the LLM's training data and answer confidence is a novel and intriguing idea which I also agree.

**Justification For Why Not Higher Score:**

RepARE as presented involves multiple LLM inference stages and off-the-shelf tools, potentially impacting its reproducibility and application feasibility due to computational cost and complexity. Some aspects of the methodology, like the oracle implementation and the choice of question candidates, require further explanation and justification. The paper could also benefit from improved clarity and organization, particularly in Section 3 on the methodology.

**Justification For Why Not Lower Score:**

The paper presents an interesting unique idea and provides a comprehensive evaluation across diverse data and analysis of different components' contributions.

The authors promise to release code and data, which is crucial for reproducibility and adoption, unlike other closed-sourced similar efforts.

---

### Decision · Program_Chairs · 2024-01-16

Accept (poster)